# CAR-T: What Is Next?

**DOI:** 10.3390/cancers15030663

**Published:** 2023-01-21

**Authors:** Yi-Ju Chen, Bams Abila, Yasser Mostafa Kamel

**Affiliations:** School of Cancer & Pharmaceutical Sciences, Faculty of Life & Sciences & Medicine, King’s College, London SE1 9NH, UK

**Keywords:** chimeric antigen receptor-t, CAR-T therapy, CAR-natural killer, CAR-NK, CAR macrophages (CAR-Ms)

## Abstract

**Simple Summary:**

In 2017, two chimeric antigen receptor-T (CAR-T) therapies were approved by the FDA for advanced/resistant lymphoma and acute lymphoblastic leukemia. However, despite the breakthrough efficacy results, the safety of CAR-T treatment is still a concern for treating physicians and their patients. Moreover, the high rate of relapse in up to 60% of patients previously treated with CAR-T represents a major challenge. There is currently extensive research activity aimed at addressing these shortfalls; strategies include changing the administration plans of CAR-T, combining it with chemotherapy, and even developing new types of CAR-T therapies. This article will focus on new CAR-T strategies that are under investigation and the results of their studies.

**Abstract:**

The year 2017 was marked by the Food and Drug Administration (FDA) approval of the first two chimeric antigen receptor-T (CAR-T) therapies. The approved indications were for the treatment of relapsed or refractory diffuse large B-cell lymphoma (DLBCL) and for the treatment of patients up to 25 years of age with acute lymphoblastic leukemia (ALL) that is refractory or in a second or later relapse. Since then, extensive research activities have been ongoing globally on different hematologic and solid tumors to assess the safety and efficacy of CAR-T therapy for these diseases. Limitations to CAR-T therapy became apparent from, e.g., the relapse in up to 60% of patients and certain side effects such as cytokine release syndrome (CRS). This led to extensive clinical activities aimed at overcoming these obstacles, so that the use of CAR-T therapy can be expanded. Attempts to improve on efficacy and safety include changing the CAR-T administration schedule, combining it with chemotherapy, and the development of next-generation CAR-T therapies, e.g., through the use of CAR-natural killer (CAR-NK) and CAR macrophages (CAR-Ms). This review will focus on new CAR-T treatment strategies in hematologic malignancies, clinical trials aimed at improving efficacy and addressing side effects, the challenges that CAR-T therapy faces in solid tumors, and the ongoing research aimed at overcoming these challenges.

## 1. Introduction

CAR-T cells are genetically modified T cells that express synthetic receptors on the cell surface to detect and eradicate cancer cells by identifying specific tumor antigens (Figure 1). Unlike T cells, CAR-T cells can recognize antigens on the surface of cancer cells without human major histocompatibility complex (MHC) molecules. Therefore, CAR-T cells can distinguish a wider range of targets than natural T cells. When CAR-T cells bind to the targeted antigen, they are activated and act as “active drugs” that attack the tumor [1]. Six CAR-T cell therapies have been granted approval for hematological cancers by the Food and Drug Administration (FDA) since 2017. Kymriah (tisagenlecleucel) and Yescarta (axicabtagene ciloleucel) were the first two CAR-T products to be approved for the treatment of patients up to 25 years of age with refractory/relapsed B-cell precursor acute lymphoblastic leukemia (ALL) and adult patients with relapsed or refractory diffuse large B-cell lymphoma (DLBCL) after two or more lines of systemic therapies [2,3]. More CAR-T cell treatments were granted approval afterwards. In total, four out of six CAR-T cell products are anti-Cluster of Differentiation (CD)19 CARs, and the two newest CAR-T cell products target B-cell maturation antigen (BCMA). [2,3,4,5,6,7]. Approved indications and details of each CAR-T cell product are shown in Table 1.

The high response rate achieved with CAR-T treatment for the above-mentioned indications was clinically meaningful in those difficult to treat patient populations with no other alternative treatment [8]. In the past few years, CAR-T cell therapies have reached considerable success in treating such blood malignancies. However, the efficacy concern that became significant is the high relapse rate after achieving remission. Likewise, the safety concern was another problem for the patients as well as treating physicians, e.g., CRS and neurologic toxicities. Apart from the seriousness of CAR-T side effects on the patients, its management added considerably to the high cost of CAR-T therapy. The main toxicity of the treatment with CAR-T therapy is cytokine release syndrome (CRS), which includes a set of life-threatening and sometimes fatal reactions after the infusion of CAR-T. Treatment with CAR-T activates T cells, leading to the release of high levels of cytokines such as IL-6, IL-10, and interferon (IFN)-Υ. The key cytokine for CRS is IL-6, which is the main cause of the immune reactions of CRS such as fever, chills, headache, and malaise. Likewise, it is also responsible for many severe and potentially life-threatening symptoms such as vascular leakage syndrome and disseminated intravascular coagulation (DIC). Neurotoxicity is another safety issue that has been frequently seen in patients receiving CAR-T cells. Although the exact mechanism of neurotoxicity is less established, probable pathophysiologic reasons have been suggested: one is that elevated levels of cytokines in the central nervous system (CNS) can lead to neurotoxicity; the other is that elevated levels of cytokines increase the permeability of the blood–brain barrier (BBB), allowing CAR-T cells to penetrate and cause neurotoxicity [9].

Management of CRS necessitates the proper assessment of its severity, usually using the CRS severity scale developed by the National Cancer Institute (NCI). This is followed by general management that includes the immediate administration of tocilizumab, which is an IL-6 antagonist given with or without corticosteroids. Additionally, specific treatment addressing the body systems and organs affected by CRS should start immediately [10]. Cytokine release syndrome (CRS) can affect different body systems and organs, with symptoms ranging from mild to severe, even debilitating. The proper management of CRS includes other medical specialties such as neurology, nephrology, and cardiology, in addition to a treating hematologist, all working together to address the different body systems/organs [11,12]. The treatment of severe CRS is usually done in an intensive care unit. The success of CRS treatment is dependent on the immediate implementation of the necessary measures upon the appearance of symptoms [13,14,15]. In the management of neurotoxicity, the standard treatment is corticosteroids due to the limited BBB penetration of tocilizumab [16].

In addition to concerns about efficacy and safety, another question is how the barriers that limit the application of CAR-T cell therapy can be overcome. About 90% of cancers are solid tumors, but current indications for CAR-T cell therapy are limited to treating hematological malignancies. The above issues suggest that there are still various obstacles to this therapy that need to be addressed. This review will demonstrate the challenges of CAR-T cell therapy in cancer treatment and propose potential strategies to optimize CAR cell therapy. This article will also provide speculation on the future development of this therapy.

## 2. Generations of CARs

The positive results of CAR-T therapy led to continued research trying to improve on the CAR-T activity while minimizing toxicity. Currently, there are four generations of CAR-T that are either developed or under development. The constitution of general CARs is demonstrated in Figure 2. The first-generation CARs were simple in their design. They only contained an activation domain (CD3ζ chains) in the intracellular region. This simple structure of CD3ζ had limited therapeutic activity. The structure can send signals leading to the activation of T cells; however, it did not support cell expansion, which is a necessary step for disease control. To overcome this issue, the second-generation CAR-T had a costimulatory signaling domain (e.g., CD28, 4-1BB, and OX40), which was added to the intracellular region resulting in a dual-signal configuration leading to significant cell expansion and improved activity [1]. The third-generation CARs (Figure 2) were developed based on the second generation. It has two costimulatory domains in addition to the activation domain CD3ζ chains in the intracellular region, thus enhancing the survival of the CAR-T cells. Subsequently, the fourth-generation CAR-T cells were developed. They are known as T cells redirected for antigen-unrestricted cytokine-initiated killing (TRUCKs). They are modified by introducing additional transgenes leading to the production of inducible cytokines (e.g., interleukin (IL)-12). This resulted in the improvement of the cell function in addition to the modulation of the tumor microenvironment (TME) [17].

## 3. Efficacy Issues and Strategies to Overcome

### 3.1. Mechanisms of Relapse to CAR-T Cell Therapy

Patients who relapsed following CAR-T therapy fall into two main groups: antigen-positive relapse and antigen-negative relapse [11]. An antigen-positive relapse occurs when the response to CAR-T therapy is not sufficient, with a minimal effect for CAR-T cells and only transient B-cell aplasia. With flow cytometry, CD19 can be detected on the surface of the cells [18]. This type of relapse is associated with poor CAR-T cell persistence and proliferation in the human body. An antigen-negative relapse is caused by antigen escape, which may be caused by a gene mutation. For instance, the CD19-negative relapses represent up to 20% of the relapsed CAR-T cell-treated patients. With a CD19-negative relapse, CD19 is not detected on the surface of the cells [19]. This occurs due to mutations in exons 1–13 of the CD19 gene. Exons 1–4 carry the codes for the extracellular domains, and exons 5–13 carry those of the transmembrane domain [20,21]. The percentage of CD19-negative cells can be measured by flow cytometry, which is also able to assess the frequency of allelic mutations and the percentage of cells where there is a homozygous loss and biallelic mutations, which are major reasons for the loss of the targeted epitope of CD19, with the subsequent escape from the effect of anti-CD19 CAR-T [21].

### 3.2. Strategies to Overcome CAR-T Relapse

Relapse after initial response to CAR-T cells is still a major problem. Different strategies were developed to overcome this issue, e.g., the development of novel CAR-T designs and combinations with other therapies. Some of these strategies are summarized in Table 2.

### 3.3. Strategies to Overcome Antigen-Positive Relapse

#### 3.3.1. Combination with Other Therapy

To overcome this type of relapse, ongoing research is evaluating the combination of CAR-T with other therapies such as the development of artificial antigen-presenting cells (AAPCs) and the development of CARs with novel designs. AAPCs are a technology aimed at developing a platform in which the APC provides controlled signals to stimulate the T-cell expansion and specific molecules on the T cells with a specific molecular phenotype and activity. The AAPCs can be derived using signal transduction of specific cell lines or from synthetic materials such as polystyrene coated with specific cytokines or other stimulatory molecules that will deliver the required signal to the T cells. Giving AAPCs to regularly stimulate CAR-T cells after disease remission is a novel strategy for prolonging clinical response. It is currently undergoing testing in a clinical study [22,23,24]. Additionally, research on cancer immunology has shown that the signals sent to the T cells by professional antigen-presenting cells at the time of the encounter of tumor antigen can affect the efficacy of the treatment of CAR-T cells. This has resulted in the activities aiming at developing artificial antigen-presenting cells in or to control the signals sent to the T cells when tumor antigen is encountered.

#### 3.3.2. Novel CAR Designs

Another attempt to prevent antigen-positive cells is the generation of CAR-T cells that have a truncated IL-2 receptor β-chain domain with a transcription factor, e.g., a 3/5 transcription factor and a signal transducer. These are considered fifth-generation CARs. This concept is currently in the exploratory stage; however, it has shown promising activities when tested in preclinical models of different blood and solid cancers by showing improved cell proliferation and persistence in comparison with the currently available CAR-T cells [25,26].

### 3.4. Strategies to Overcome Antigen-Negative Relapse 

#### 3.4.1. Combination with Other Therapy

Another approach to preventing antigen-negative relapse is combining bi-specific T-cell engagers (BiTEs) with CAR-T cell therapy. One side of a BiTE is designed to bind to CD3 and activate T cells, while the other side is directed against a tumor antigen. The combination with BiTEs allows CAR-T cells to aim at distinct antigens by altering BiTEs. This concept has been tested in preclinical models to avoid antigen escape. The combination of BiTEs and CAR-T is a bispecific antibody (BsAbs) that recruits T cells to tumor cells. Both are effective immunotherapies; they redirect T cells to a tumor-specific antigen using antibody fragments. They have shown considerable success in B-cell hematologic malignancies and are currently tested for the treatment of solid tumors [27]. A preclinical study has shown that CAR-T.BiTE, the combination of CAR-T cells with BiTEs, can eliminate tumors with a heterogeneous EGFRvIII expression in a mouse model. EGFRvIII is a specific tumor antigen of glioblastoma (GBM), while EGFR is an antigen often overexpressed in glioblastoma but also expressed in normal tissues. When the EGFRvIII mutation is lost in GBM, the wild-type EGFR is still maintained. Therefore, the CARs on T cells are designed to attack tumor cells that express EGFRvIII. Meanwhile, CAR-T cells secrete BiTEs to redirect CAR-T cells and recruit bystander T cells against wild-type EGFR. The elimination of the EGFRvIII-negative, EGFR-positive glioblastoma has been seen in a study on treatment with CART.BiTE [28].

#### 3.4.2. Novel CAR Designs

Multiantigen-specific CARs with “OR” logic gates target distinct antigens on the same T-cell surface. The design is to present bispecific CARs on a single T cell to target two different antigens expressed on the tumor. It has been tested in preclinical research and has shown positive results. For instance, multiple trials have reported patient relapses due to the appearance of CD19-negative leukemic cells. In a preclinical study, the mixed (75% CD19+/CD20+, 25% CD19−) B-cell malignancies in mice were completely cleared after receiving the CD19/CD20 OR-gate CAR-T cells and did not show signs of relapse over 109 days. However, single-input CD19 CAR-T cells can only eliminate purely wild-type (CD19+/CD20+) tumor cells but not mixed B-cell tumors [29]. An alternative to the “OR” logic gates strategy is a pooled CAR-T cell that uses two CAR-T cell lines that target more than one antigen. The engineered human T cells are activated by a signal from one antigen, while the co-stimulation signaling function is carried by a co-stimulation that is specific to the second antigen. This concept has been proven pre-clinically and is currently in the clinical phase of research [30,31,32]. Universal CAR development enables a single line of CAR-T cells to bind to several antigens by giving different adaptor molecules as ligands. This approach is aimed at overcoming the antigen escape problem without the need to manufacture additional CAR-T cells [33,34].

## 4. CAR-T Therapy Side Effects and Strategies to Overcome Them

### 4.1. On-Target, On-Tumor Activity: Cytokine-Related Toxicity

The extent of cytokine release reflects the degree of T-cell activation, which is crucial to achieving sufficient clinical effects. Nevertheless, it is also the cause of immune-related adverse events (irAEs), which includes CRS and neurotoxicity. Therefore, the strategy for preventing cytokine-related toxicity is to either diminish T-cell activity or block cytokine effects. 

Recently, a panel of experts that included oncologists, neurologists, cardiologists, and emergency medicine specialists reviewed the available studies for CAR-T and the irAEs developing as a result of the treatment with CAR-T. The outcome of the discussion was published as the American Society of Clinical Oncology (ASCO) guidelines for the management of irAEs in patients treated with CAR-T therapy. The guidelines are intended to support the treating physician in managing the most common irAEs occurring during treatment with CAR-T, which includes CRS, immune effector cell-associated neurotoxicity syndrome (ICANS), infections, B-cell aplasia, and cytopenia. Management of CRS was mentioned above. Other serious irAEs that can occur are neurological toxicities, which are also known as ICANS. They usually occur after four days of therapy [35]. ICANS is the second serious irAE that is not infrequent in patients treated with CAR-T cells. The patients suffer from encephalopathy with other different symptoms, e.g., behavioral changes, aphasia, fine motor impairment, and headache [36]. Severe cases might also require admission to the ICU in order to control seizures and intubating the patient if needed to avoid any damage to the airway passages. ICANS can occur with CRS or alone, and it can occur up to 1 month after the treatment with CAR-T. It can be self-limited, with symptoms resolving by 17 days, or may become severe and cause permanent neurological damage. Treatment of ICANS includes the administration of corticosteroids and supportive care. Tocilizumab is contraindicated for the treatment of neurotoxicity, as it can worsen the symptoms. As CRS can occur with ICANS and CRS necessitates the treatment of tocilizumab, in accordance with ICANS, if it is low-grade and spontaneously resolves, the treatment of CRS can start and tocilizumab can be used. If the ICANS is severe and requires active measures, then managing ICANS is the first priority, before management of the CRS [36,37]. 

### 4.2. Strategies to Minimize Cytokine-Related Toxicity

Research and clinical activity are continuously trying to improve on the outcome of treatment with CAR-T while minimizing toxicity. New concepts, combinations, and designs were made to achieve these objectives, and some of these activities include the following: 

#### 4.2.1. Combination with Other Therapy

Based on preclinical studies, the administration of IL-1 and catecholamines antagonists at the time of CAR-T infusion have shown a decrease in the occurrence of CRS and have minimized its severity. Ongoing clinical studies are probing this concept while also assessing the impact on efficacy [35,36]. Dasatinib is a tyrosine kinase that is approved for the treatment of Philadelphia chromosome-positive chronic myeloid leukemia and ALL. Mestermann et al. reported that dasatinib has a reversible inhibitory effect on lymphocyte-specific protein tyrosine kinase (LCK) in the CD3ζ domain. When dasatinib was administered three hours after CAR-T therapy infusion, it markedly reduced the occurrence of death due to CRS in lymphoma mouse models. Clinical studies are ongoing to assess the ability of dasatinib to decrease CRS following the infusion of CAR-T without eradicating CAR-T cells that could decrease the efficacy of the CAR-T therapy [38,39].

#### 4.2.2. Novel CAR Designs 

##### ON-Switch CARs and OFF-Switch CARs

Engineered CAR-T cells, once infused into patients, start a chain of immunological reactions that kill the target cells. However, these reactions continue in an uncontrolled way, leading to a set of immunological toxicities leading to the development of different symptoms of CRS. The release of these inflammatory cytokines in an uncontrolled manner impacts different organs with different severities. The ON-switch CAR is a design that separates the signal domain from the costimulatory domain. The T-cell activation can occur by adding heterodimerizing small molecules that promote the assembly of two fragmented CARs. Furthermore, the extent of cell activation can be modified by the dosage of the molecules [39]. Another design to reduce toxicity is the OFF-switch CAR, which is also known as the small molecule-assisted shutoff (SMASh)-CAR. SMASh-CAR includes a degron domain in the CAR structure and enables CAR degradation when protease inhibitors are given, thus downregulating T-cell activity. Oppositely, without the administration of a protease inhibitor, protease would cut off the target site to remove degron and lead to the expression of CAR [40].

##### Suicide Gene/Receptors

CAR-T therapy is currently the most effective therapy for its approved indications, but the cellular killing ability of the cells is not restricted to cancer cells only. It can affect normal cells leading to off-target effects and different toxicities. To minimize these toxicities, a genetically encoded molecule aiming at the selective destruction of the transferred cells was developed. This is called a suicide gene. The addition of a suicide gene with CAR-T cells leads to the selective destruction of the adoptively transferred cells as well as a selective ablation of the genetically modified cells, preventing the off-target effect of CAR-T therapy and the toxicity of the therapy. There are several validated suicide genes, two of which are among the most studied in the clinical setting: herpes-simplex-thymidine-kinase (HSV-TK) and inducible-caspase-9 (iCasp9). The iCasp9 suicide gene can trigger apoptosis rapidly when exposed to dimerizing agents (AP1903), thereby leading to an irreversible termination of T cells, and the inducible HSV-TK suicide gene is activated by the administration of ganciclovir (GCV) [41]. Similarly, the strategy of suicide receptors was pursued; it depends on inducing another antigen (e.g., CD20) expression on the surface of CAR-T cells, thereby serving as a “suicide receptor” that, when anti-CD20/rituximab is infused, will bind to the receptors and eliminate CAR-T cells [39,40,42].

### 4.3. On-Target, Off-Tumor Activity 

An ideal target for CAR-T should only exist on cancer cells and be expressed at a high level. Unfortunately, most antigens are present on both tumor cells and normal cells, resulting in CAR-T cell therapy that targets antigens on both normal and cancer cells. When CAR-T cells attack antigens that coexist in some normal tissues, this is what is called “on-target, off-tumor” toxicity. In patients who received anti-CD19 CARs, an eradication of normal B cells has been observed, since the B-cell line presents CD19 on their surface. Thankfully, this negative effect was countered to some extent by the infusion of gamma globulin. However, on-target, off-tumor toxicity could lead to severe side effects such as multiorgan dysfunction and even CAR-related death under severe circumstances [43]. Some novel CARs have been developed to minimize the off-target activity by enhancing tumor specificity. To achieve this, the researchers targeted two antigens, with co-expression being specific to the tumor [30,31]. Further details are demonstrated below.

### 4.4. Novel CAR Designs to Avoid On-Target, Off-Tumor Activity

In mice studies, CAR-T cells targeting tumors with ROR1 antigen following lymphodepletion resulted in lethal bone marrow failure. This is due to the targeting of ROR1 expressed on normal stromal cells. The ability to develop a CAR-T that selectively targets antigens present on tumor cells and not normal cells will improve the efficacy results but avoid severe toxicities—the so-called gated or logic-gated approach. In pre-clinical studies, synNotch receptors induced transcriptional activation in response to combinatorial target–antigen recognition. However, data from clinical studies are needed to define the applicability of different strategies in clinical practice [32].

#### 4.4.1. Dual CARs with a “NOT” Logic Gate

The signal domain of iCAR is originated from immune checkpoints, such as cytotoxic T lymphocyte-associated antigen 4 (CTLA-4) and programmed cell death protein 1 (PD-1). CTLA-4 and PD-1 are immunosuppressive receptors commonly found on regulatory T lymphocytes (Tregs) and other immune cells. When CTLA-4 and PD-1 are activated via binding to corresponding antigens, inhibitory signals will be transduced to decrease cytokine secretion and suppress immune cell activation. Consequently, when iCARs detect antigens on normal cells, they will inhibit the activation of CAR-T cells and serve as a dynamic controller to reduce on-target, off-tumor toxicity [44].

#### 4.4.2. Dual CARs with an “AND” Logic Gate

The synthetic Notch (SynNotch) system is an effective method for enhancing the on-target activity of CAR-T cells by the design of dual CARs with “AND” logic gates. When the CAR on the cell surface recognizes the target antigen, it triggers transcription and synthesizes another CAR that recognizes a different antigen. In this case, CAR-T cells are activated and attack the tumor only when the dual tumor antigens are present, thereby minimizing off-target tumor activity [32]. Combination CAR, also known as split CAR, is another strategy employed to diminish off-tumor toxicity. The signaling and co-stimulatory domains are split into two CARs that recognize distinct tumor antigens. Therefore, to activate CAR-T cells, both target molecules must be present at the same time [34].

## 5. Challenges of CAR-T Cells in Solid Tumors and Strategies to Overcome Them

Although solid tumors are the most numerous of all cancers, CAR-T cell therapy is still not applicable to solid tumors. There are several barriers to this therapy in the treatment of solid tumors, including antigen specificity/heterogeneity, poor cell trafficking, and the immunosuppressive tumor microenvironment (TME).

### 5.1. Antigen Specificity/Heterogeneity

The lack of antigen specificity and antigen heterogeneity are important obstacles for CAR-T cells in treating solid tumors, leading to severe on-target, off-tumor toxicities, and insufficient efficacy. The ideal targets for CAR-T cells are tumor-specific antigens (TSAs), which are expressed at high levels on tumor cells, and healthy tissues would thus not be damaged. CD19, CD22, and BCMA, which are highly restricted to the B-cell lineage antibodies, are currently known to be close to TSAs. The most common on-target, off-tumor toxicity caused by CAR-T cells targeting CD19, CD22, and BCMA is profound B cell aplasia, but this could be treated with intravenous immunoglobulin (IVIG) replacement therapy. Unfortunately, antigens are expressed on the surface of solid tumors, and most of the hematological malignancies are tumor-associated antigens (TAAs), which also exist on normal “bystander” cells [9]. Consequently, less tolerable or even life-threatening toxicities could be caused by CAR-T-cell therapy. Furthermore, antigen heterogeneity results in various levels of antigen expression in different tumor sites, which impede the ability of CAR-T cells to detect tumors [43].

Despite the efforts of researchers, finding optimal antigens in solid tumors is not easy. The development of novel CARs may be a more feasible approach to overcoming the antigen specificity/heterogeneity barrier. Combining BiTEs with CAR-T cells could be one way to tackle the problem of antigen heterogenicity in solid tumors. As mentioned in previous paragraphs, this strategy allows CAR-T cells to recognize different antigens by altering BiTEs. It has been tested in several solid tumor models in preclinical studies and has shown positive results [45]. Some novel CARs such as the SynNotch receptor system and combination CARs, which are used to reduce on-target, off-tumor activity, can also promote antigen coverage and increase antigen selectivity in solid tumors and have demonstrated positive clinical outcomes in preclinical studies [32,38]. Universal CAR, previously mentioned in regard to overcoming antigen-negative relapse, is another way to target multiple antigens to overcome antigen heterogenicity [46].

### 5.2. Cell Trafficking and Infiltration

CAR-T cells can only be activated when they detect and bind to antigens expressed on the surface of tumor cells. For hematological malignancies, tumor cells are blood cells in blood vessels. In the treatment of solid tumors, CAR-T cells need to penetrate the vascular endothelium and enter solid tumor neoplastic lesions. The success of trafficking mainly depends on the proper pairing between chemokines secreted by tumor cells and receptors on T cells (typically CXCR3 and CCR5). However, they are often mismatched with each other, or tumor cells produce too few CXCR3 and CCR5 ligands, hindering T-cell trafficking and infiltration [47]. One of the strategies to improve trafficking issues is regional delivery. CAR-T cells are directly injected into the tumor site. Another potential approach is to provide high concentrations of CAR-T cells by implanting biopolymer devices in neoplastic tissues. Some novel CARs have been designed to overcome ineffective trafficking as well. CAR-T cells can be engineered to express tumor-associated chemokine receptors to bind to corresponding chemokine ligands and improve T-cell migration. In preclinical studies, this approach has successfully attracted CAR-T cells to the melanoma site. The development of FAP-specific CAR-T cells may be an approach to overcome the physical barriers. Cancer-associated fibroblasts (CAFs) compose up to 90% of the tumor extracellular matrix, and they produce high levels of fibroblast activation protein (FAP). Therefore, the design of FAP-specific CAR-T cells could be a feasible strategy to improve cell penetration [48]. Cell infiltration can also be increased by engineering CAR-T cells to generate enzymes (e.g., heparinase) for degrading and modifying the extracellular matrix [49]. However, the efficacy and toxicity of these strategies are undetermined, and further investigation is needed.

### 5.3. The Tumor Microenvironment

The tumor microenvironment (TME) is a unique environment created by tumor cells in which low pH, hypoxia, and nutrient deficiency are observed. Moreover, TME tends to have more inhibitory soluble cytokines, inhibitory immune cells, and inhibitory immune checkpoints (e.g., PD-1 and CTLA-1) than normal tissues, hindering T-cell activity [47,50,51]. To decrease the amount of suppressive immune cells, using a blocking antibody is one of the strategies employed to remodel the TME. However, preconditioning chemotherapy may not be the most ideal way to achieve this since it will also suppress T-cell activity. A better strategy is to use an antibody that only targets immunosuppressive cells, such as the anti-regulatory T cell (Tregs) antibody and the anti-myeloid-derived suppressor cells (MDSCs) antibody [52,53]. Likewise, a blocking antibody can also be used to lower the level of inhibitory cytokines, especially primary ones in the TME, such as transforming growth factor-β (TGF-β). An alternative option is to knock out the TGF-β receptor on T cells via gene editing [54].

Immune checkpoints are safe regulators of the immune system, preventing an overly strong immune response from damaging normal cells in the body. However, some tumor cells can upregulate the number of immune checkpoints on T cells and stimulate them to constrain T lymphocyte activity by secreting their ligands. Therefore, the combination of CAR-T cells with immune checkpoint inhibitors (ICIs) may be a strategy to enhance T cell-mediated toxicity. Currently, there are several ICIs available on the market, such as CTLA-4 inhibitors and PD-1/PD-L1 inhibitors [55]. An alternative strategy is to knockout PD-1 expression of CAR-T cells by gene modification. Other novel CAR designs such as armored CAR-T cells and the fourth-generation CAR-T cells (TRUCKs) are genetically engineered CARs to produce cytokines, which can stimulate an immune reaction and modulate the TME. Intrinsic resistance to Tregs was seen in a mouse model after infusing “armored CAR-T cells,” which secreted IL-12, a pro-inflammatory cytokine that can activate immune cells [56]. Fourth-generation CARs differ from armored CAR-T cells in that they are more flexible in design. They begin to synthesize stimulatory cytokines only when the target antigen is detected. Fourth-generation CARs have been tested in preclinical models of various solid tumors and have shown remarkable efficacy [57].

### 5.4. CAR-T Studies on Solid Tumors

CAR-T therapy was assessed for the treatment of different solid tumor types, e.g., colorectal cancer (CRC), breast cancer (BC), thoracic tumors, hepatocellular carcinoma (HCC), and ovarian cancer. This review will focus on four of the common tumor types and the trial of CAR-T therapy in its management. In CEA-positive CRC: a CAR-T was specifically developed for the carcinoembryonic antigen (CEA), which is a tumor marker expressed in the majority of CRC. Currently, there are several ongoing Phase I studies evaluating the safety and efficacy of CEA-directed CAR-T therapy in advanced CRC. The results of these studies are still awaited. Minimal data come from a case report of a patient with CRC who had a complete metabolic response within the liver lasting for 13 months, and preliminary results of a phase I study show that, out of 15 patients with unresectable metastatic CRC who received CAR-T therapy, there were 2 partial responses, and 9 achieved stable disease [58]. In a phase 1 trial, prostatic cancer patients were treated with lymphodepletive chemotherapy, which was followed by CAR-T therapy, received as a continuous infusion with an escalating dosage. The treatment was tolerable, and 2 out of 5 patients (40%) achieved Partial Remission (PR) and a Prostate-Specific Antigen (PSA) decline of 50–70%. Another patient showed a minor response to treatment [59].

In hepatocellular carcinoma (HCC), a CAR-T was developed against a potential antigen target, i.e., glypican-3 (GPC3). In two phase I studies assessing the effect of CAR-T therapy on GPC3+ HCC, patients were given an infusion of cyclophosphamide and fludarabine-based lymphodepletion. There were 13 patients enrolled in the study, with 9 of the 13 patients (69%) suffering from CRS. There were no grade 3/4 neurotoxicities. The Overall Survival (OS) was 50.3% at 6 months, 42% at 1 year, and 10.5% at 3 years. Of note, one patient from the study maintained stable disease and survived for 44 months. In another phase I study, patients with advanced metastatic solid tumors were treated with CD133 targeting CAR-T. The study enrolled patients with different solid tumors, among whom were 23 patients with HCC. Three patients achieved PR (13%), and 14 patients achieved stable disease (SD) (61%). The median Progression-Free Survival (PFS) was 5 months, and the 3-month disease control was 65% [59,60]. For the treatment of thoracic cancer, a wide variety of targets, including EGFR, HER2, MSLN, MUC1, CEA, ROR1, and PD-L1, are currently being evaluated for CAR-T cell therapy in lung cancer. Among these, EGFR- and MSLN-specific CAR-T cells seem to be more promising than the others due to the antigen’s higher specificity and lower on-target, off-tumor toxicity concerns.

An open-label phase I investigated the use of regionally delivered autologous mesothelin-targeted CAR-T with pembrolizumab for malignant pleural mesothelioma (MPM). Pembrolizumab was given based on pre-clinical work that has shown that the PD-1 blockade can enhance mesothelin CAR-T activity and rescue the function of exhausted CAR-T cells [61]. In this study, 27 patients with malignant pleural disease (either as primary or pleural metastases) received intrapleural mesothelin targeting CAR-T; 25 of these patients had a diagnosis of malignant pleural mesothelioma (MPM), and 18 of them received pembrolizumab after CAR-T cell therapy. The median overall results were based on 23 out of the 27 patients in the intrapleural mesothelin targeting CAR-T and then the 18 patients who received both pembrolizumab and CAR-T therapy. At a median follow-up of 20 months, the CAR-T treatment median OS was 17.7 months, and the 1-year survival rate was 74%. Pembrolizumab was given every 3 weeks to promote CAR-T function. With the addition of pembrolizumab, the patients’ results were further enhanced with a median OS of 23.9 months, and a 1 year-survival rate of 83% [62,63,64].

## 6. CAR-Based Cellular Therapies in the Future: Off-the-Shelf CARs and Next-Generation CARs

Autologous CAR-T cells have shown remarkable clinical outcomes and have dramatically changed the treatment of blood cancers. However, there are still issues that prevent patients from receiving CAR-T cell therapy. In addition to the efficacy and safety issues of CAR-T therapy mentioned in the previous sections, the high cost, complex process, and rather long waiting time of around 3 weeks needed for manufacturing personalized T cells are also factors that hinder patients’ access to treatment [65]. Consequently, to overcome these obstacles, the development of universal allogeneic CAR-T cells (also known as “off-the-shelf” CAR-T cells) and other CARs using alternative effector cells are underway.

### 6.1. Off-the-Shelf CAR-T Cells

Compared to autologous CAR-T cells, off-the-shelf CARs have numerous potential benefits. Firstly, healthy donors can be selected and used as a source of immune cells to generate them. This could lead to a better performance of CAR-T cells since the immune cells from healthy donors have not been impacted by cancer effects or by exposure to chemotherapeutic agents. Some clinical and preclinical research has shown that the efficacy of CAR therapy is affected by the quality and quantity of T cells retrieved from patients [66]. Furthermore, patients who have received chemotherapy before the infusion are more likely to have poor-quality T cells, as well as an insufficient numbers of them, than those who have not, and this could be related to poor CAR-T-cell activity and failing to harvest enough cells for CAR-T manufacturing [67]. Secondly, unlike the time-consuming process of producing personalized autologous CAR-T cells, large numbers of allogeneic CAR-T cells can be generated from a single donor, producing batches of preserved CAR-T products that can provide patients with immediate access to treatment. 

In addition, the price of CAR-T therapy can be significantly reduced through large-scale production processes [65]. Nevertheless, it is concerning that, if the immune cells are originated from MHC-mismatched donors, off-the-shelf CAR-T products could result in graft rejection and graft-versus-host disease (GVHD), which could be life-threatening. GVHD is a disadvantageous immune reaction that happens upon administration to allogeneic CAR-T cells, and it is due to a “non-self” recognition by the host immune system. Moreover, GVHD could lead to the decreased antitumor activity of allogeneic CAR-T cells, since they could be eliminated by host immune cells. Several different sources of T cells for allogeneic approaches have been tested in preclinical and clinical studies to overcome GVHD, such as virus-specific T cells, gene-modified conventional T cells, and non-conventional T cells [68].

The use of virus-specific T cells may be a potential way of reducing the risk of GVHD, as they have long been used to treat viral infections after transplantation [69]. The safety of this approach has been shown in a phase I trial in which there were no reports of severe GVHD in patients with B-cell cancer who were given CAR virus-specific T cells [70]. T-cell receptors (TCRs) are primarily responsible for the recognition of MHC molecules between foreign substances and host cells. Therefore, the use of genetic modification to remove endogenous molecules such as αβ TCR and MHC is another potential strategy to overcome the problems of GVHD and rejection. In a clinical study, two infants with R/R ALL were successfully treated with universal CAR-T cells generated by transcription activator-like effector nuclease (TALEN) gene engineering [71]. The introduction of CARs into various T-cell subtypes is currently under investigation. Gamma delta T cells (γδ T cells), which account for around 5–10% of the T-cell population, are one of the most promising candidates for off-the-shelf CAR production, as γδ TCR expressed on the surface of γδ T cell is MHC-independent, thereby reducing the risk of GVHD [72]. In addition, the promises of CAR-γδT cells in treating solid tumors have been demonstrated in a proof-of-concept study, where additional antitumor activity was seen in CAR-γδT cells, while the intrinsic γδT function still exists [73].

### 6.2. Next-Generation CAR Cells

Despite the great success of CAR-T therapy in hematological malignancies, its toxicity and the other limitations mentioned above have significantly hindered patient access and its expansion in solid tumor treatment. Given these deficiencies, there is growing interest and substantial research into finding alternative effector cells for CAR cell therapy. NK cells and macrophages are two promising candidates for the manufacture of next-generation CARs because of their favorable properties. Unlike T cells, they are members of the innate immune system that can directly identify target cells without MHC and do not cause GVHD. Therefore, they are also possible options for producing off-the-shelf CARs. 

NK cells have several advantages over CAR-T cells, one of which is that they can recognize tumor cells even when MHC molecules are downregulated and allow them to avoid antigen escape [74]. Furthermore, NK cells can be extracted from various allogeneic sources, such as induced pluripotent stem cells (iPSCs) or umbilical cord blood, as NK cell activation does not need to go through the MHC pathway. The antitumor activity of CAR-NKs has been demonstrated in preclinical models of various hematological and solid tumors [75]. Moreover, in a phase I/II clinical study enrolling 11 patients with CD-19-positive hematological malignancies, no significant toxicity was reported after the administration of allogeneic CAR-NK cells [15]. CAR-NK-based clinical studies that have been registered on clinicaltrials.gov and are active or recruiting are demonstrated in Table 3 [76].

Macrophages represent another promising candidate that has unique features. First, they can attack tumor cells via selective phagocytosis and present antigens to T cells for adaptive immunity activation. Second, macrophages are the amplest and most highly infiltrated innate cells in the tumor microenvironment. Finally, they can generate chemokines or cytokines and thus serve as major immunomodulators to remodel the suppressive tumor microenvironment [75]. In preclinical studies, CAR macrophages successfully destroyed cancer cells in vitro and reduced tumor burden in mouse models with two different solid tumors, resulting in improved OS. Other research showed that CAR macrophages promoted the secretion of pro-inflammatory cytokines and enhanced T-cell cytotoxic activity against tumors [77]. Notably, a CAR-M-based phase I clinical study has been registered on clinicaltrials.gov and has started recruiting (Table 4) [78].

## 7. Discussion: Current Challenges of CAR-T Cell Therapy and Future Development Activity

### 7.1. Target Antigen Selection

More research has been devoted to developing such therapies due to their excellent clinical results. However, CD19 remains the most popular target antigen since Kymriah was approved as the first anti-CD19 CAR product in 2017. Even though the expression patterns of a few targets, such as CD20, CD22 and BCMA, are similar to that of CD19, they have not been used as successful as CD19. One probable reason is that CD19 expression is more stable than others, leading to the stronger antitumor activity of anti-CD19 CAR in clinical implementation [79]. The recent approval of the first non-CD19-directed CAR Abecma in 2021 highlights the difficulty of finding ideal targets for CAR-T cell therapy. Encouragingly, the second non-CD19-directed CAR Carvykti was then approved in the following year. Abecma and Carvykti are both anti-BCMA CAR-T cells.

### 7.2. Efficacy, Safety, and Clinical Application Extensions

CAR-T cell therapy has revolutionized the treatment of several blood cancers that once had no appropriate treatment options. However, given this urgent unmet medical need, the approval of CAR-T cell therapy was based on a considerably short study duration, weaker primary endpoints, and a limited number of patients compared to traditional oncology trials. CAR-T cell products currently on the market are conducted in phase I/II or phase II trials with open-label and single-arm experimental designs. Therefore, some drawbacks of the design have been noted. First, these trials had less than 1000 subjects and a follow-up duration shorter than two years. Next, the primary endpoint of most studies is the Overall Response Rate (ORR), the Complete Response (CR) rate, or the Duration of Remission (DOR), rather than a more robust endpoint such as OS or PFS. Last, the single-arm design had no control group to assess how much preconditioning chemotherapy affected patients prior to CAR-T cell infusion or how much of the antitumor response was attributable to CAR-T cell therapy. Accordingly, larger studies with time to event endpoints and a sufficient patient follow-up duration is needed.

CRS and neurologic toxicity are potentially life-threatening serious AEs and greatly restrict CAR-T cell therapy. Both remain unpredictable due to a deficient understanding of the exact mechanisms of causality. It is thought that a more informative preclinical model needs to be developed to better understand these toxicities. Moreover, each investigational site has its own grading tool and treatment guidelines for these toxicities, making data collection and safety evaluation difficult. Excitingly, ASTCT Consensus Grading was published in 2018 as a consensus grading system for CRS and neurotoxicity associated with immune effector cells [80].

It is encouraging that new CAR designs are being developed to overcome the above challenges, and some of these designs have shown positive results in preclinical or clinical studies. Dual CARs with AND gates are a promising strategy to expand antigen coverage, while dual CARs with NOT gates can reduce toxicity by acting as controllers of CAR-T cell antitumor activity. With the capability of secreting immune-modulating molecules, some innovative CARs can increase T-cell function and adjust the TME to restore T-cell antitumor activity. However, more time is needed to properly assess the efficacy and safety of these new CAR-T designs. 

### 7.3. Treatment Costs

The notable outcomes of CAR-T cell therapy were considered a clinical success; however, on the commercial side, CAR-T therapy has achieved minimal success. The cost of CAR-T cell therapy is a significant barrier to patient access due to the complicated, highly personalized, and time-consuming manufacturing procedure. For example, for a single infusion, Kymriah costs $475,000, and Yescarta costs $373,000, excluding hospitalization for treatment side effects [81]. Such a high price is a financial burden on both individuals and the healthcare system and limits the access of CAR-T to patients who need it. 

### 7.4. Future Perspectives

Although the development of CAR-T cell therapy still faces many challenges and obstacles, the continuous increase in the number of registered clinical studies indicates that the field is flourishing. New designs of CAR-T cells and novel CARs such as CAR-NK and CAR-M have shown potential for the treatment of solid tumors. The unique properties of CAR-NK and CAR-M also make them promising candidates for the development of off-the-shelf CAR products. Therefore, it is believed that CAR-based cell therapy will continue to be developed and optimized through emerging research and will benefit more cancer patients in the future.

## 8. Conclusions

CAR-T cell therapy has shown excellent clinical outcomes and has significantly transformed the treatment of various R/R hematological malignancies that previously have not had many treatment options. However, high treatment prices impose a substantial burden on patients and payers, thus hindering its commercial success. Furthermore, a high relapse rate, tumor antigen escape, and severe CAR-related toxicities are unresolved concerns. Nonetheless, the continuous development of CAR technology, novel CAR development and next-generation CARs, such as CAR-NKs and CAR-Ms, and CAR-based immunotherapy all have the potential to overcome the present restrictions and achieve a safer, more effective, and broader application in cancer treatment. Likewise, it is important that CAR-T therapy is affordable so that more patients can have access to it. This will help to increase our knowledge of the efficacy and safety of CAR-T therapy in practice.

## Figures and Tables

**Figure 1 cancers-15-00663-f001:**
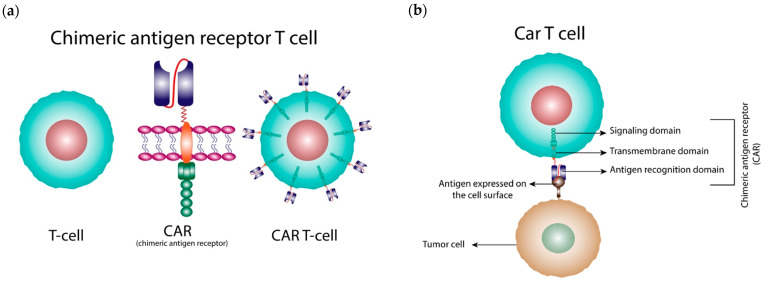
(**a**) The T cell, CAR (chimeric antigen receptor), and CAR-T cell. (**b**) Structure of the CAR-T cell and how it recognizes the tumor cell. The primary construct of a CAR consists of three parts: an antigen recognition domain, a transmembrane domain, and a signaling domain. When the antigen recognition domain of CAR binds to the antigen on the tumor cell, the CAR-T cell will be activated and serve as a ‘living drug’ that attacks and eliminates the tumor cell.

**Figure 2 cancers-15-00663-f002:**
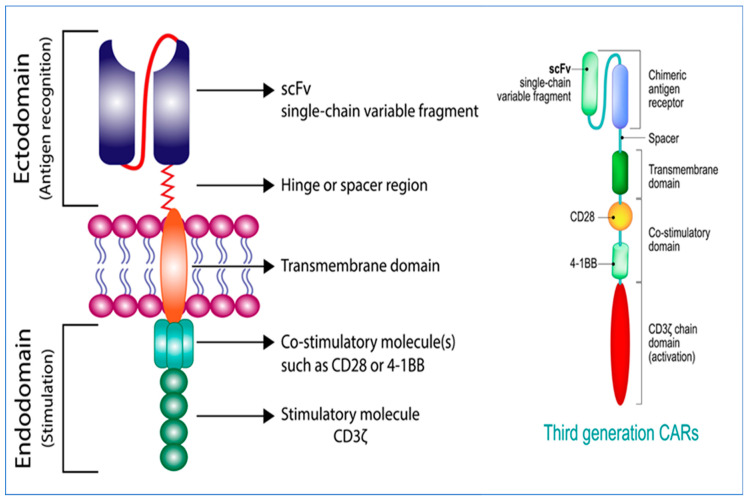
Constitution of general CARs and the third-generation CARs. Left: The primary construct of a CAR consists of three parts: an ectodomain, a transmembrane domain, and an endodomain. An ectodomain contains a single-chain variable antibody domain (scFv) to recognize tumor antigens and a spacer to provide flexibility for binding. The transmembrane region connects the ectodomain and endodomain. The endodomain is responsible for transducing signals, and it is composed of one or more costimulatory molecules, such as cluster of differentiation (CD)28 and 4-1BB, and a stimulatory molecule, CD3ζ. Right: The costimulatory domain of the third-generation CARs includes two costimulatory molecules, CD28 and 4-1BB.

**Table 1 cancers-15-00663-t001:** Indications and details of six approved CAR-T cell products [3,4,5,6,7].

CAR-T Cell Product Name and FDA Approved Date	Indication(s)	Target Antigen
**Kymriah^®^ (tisagenlecleucel)** ***-* Approved by FDA in 2017**	Patients up to 25 years of age with B-cell precursor acute lymphoblastic leukemia (ALL) that is refractory or in second or later relapse. Adult patients with relapsed or refractory (r/r) large B-cell lymphoma after two or more lines of systemic therapy, including diffuse large B-cell lymphoma (DLBCL) not otherwise specified, high grade B-cell lymphoma and DLBCL arising from follicular lymphoma. Adult patients with relapsed or refractory follicular lymphoma (FL) after two or more lines of systemic therapy.	CD19
**Yescarta^®^ (axicabtagene ciloleucel)** ***-* Approved by FDA in 2017**	Adult patients with large B-cell lymphoma that is refractory to first-line chemoimmunotherapy or that relapses within 12 months of first-line chemoimmunotherapy. Adult patients with relapsed or refractory large B-cell lymphoma after two or more lines of systemic therapy, including diffuse large B-cell lymphoma (DLBCL) not otherwise specified, primary mediastinal large B-cell lymphoma, high grade B-cell lymphoma, and DLBCL arising from follicular lymphoma. Adult patients with relapsed or refractory follicular lymphoma (FL) after two or more lines of systemic therapy.	CD19
**Tecartus^®^ (brexucabtagene autoleucal)** ***-* Approved by FDA in 2020**	Adult patients with relapsed or refractory mantle cell lymphoma (MCL). Adult patients with relapsed or refractory B-cell precursor acute lymphoblastic leukemia (ALL).	CD19
**Breyanzi^®^ (lisocabtagene maraleucel)** ***-* Approved by FDA in 2021**	Adult patients with large B-cell lymphoma (LBCL), including diffuse large B-cell lymphoma (DLBCL) not otherwise specified (including DLBCL arising from indolent lymphoma), highgrade B-cell lymphoma, primary mediastinal large B-cell lymphoma, and follicular lymphoma grade 3B, who have: refractory disease to first-line chemoimmunotherapy or relapse within 12 months of first-line chemoimmunotherapy; or refractory disease to first-line chemoimmunotherapy or relapse after first-line chemoimmunotherapy and are not eligible for hematopoietic stem cell transplantation (HSCT) due to comorbidities or age; or relapsed or refractory disease after two or more lines of systemic therapy.	CD19
**Abecma^®^ (idecabtagene vicleucel)** **- Approved by FDA in 2021**	Adult patients with relapsed or refractory multiple myeloma after four or more prior lines of therapy, including an immunomodulatory agent, a proteasome inhibitor, and an anti-CD38 monoclonal antibody.	BCMA
**Carvykti^®^ (ciltacabtagene autoleucel)** **- Approved by FDA in 2022**	Adult patients with relapsed or refractory multiple myeloma after four or more prior lines of therapy, including a proteasome inhibitor, an immunomodulatory agent, and an anti-CD38 monoclonal antibody.	BCMA

**Table 2 cancers-15-00663-t002:** Strategies to overcome the challenges of CAR-T cell therapy.

	Efficacy		Safety		Solid Tumours		
	Antigen-Positive Relapse	Antigen-Negative Relapse	Cytokine-Related Toxicity	On-Target Off-Tumor Toxicity	Antigen Specificity/Heterogeneity	Cell Trafficking	Tumor Microenvironment
**Novel CAR designs**							
The fifth-generation CARs	●						
Multiantigen CARs with “OR” logic gate		●			●		
Pooled CAR-T cells		●			●		
Universal CARs		●			●		
On-switch CARs			●				
Off-switch CARs			●				
Suicide gene			●				
Suicide receptor (Antibody-mediated depletion)			●				
CAR-T cells with tumour-associated chemokine receptors						●	
FAP-specific CARs						●	
Modify CAR-T cells to express heparinase by gene editing						●	
Dual CARs with “NOT” logic gate: inhibitory CARs (iCARs)				●			
Dual CARs with “AND” logic gate: SynNotch receptor system				●	●		
Dual CARs with “AND” logic gate: Split CARs (Combination CARs)				●	●		
Armoured CAR-T cells							●
The fourth-generation CARs (TRUCKs)							●
**Combination with other therapeutic agents**							
Artificial antigen-presenting cells (AAPCs)	●						
Bi-specific T cell engagers (BiTEs)		●			●		
Haematopoietic stem cell transplantation after remission		●					
Cytokine inhibitors			●				
Dasatinib to inhibit CD3ζ downstream signal			●				
Antibodies for depleting suppressive immune cells/cytokines							●
Immune checkpoint inhibitors (ICIs)							●
**Other strategy**							
Regional delivery				●		●	

**Table 3 cancers-15-00663-t003:** Non-comprehensive list of CAR-NK-based clinical studies which are active/recruiting.

NCT Number	Stage	Status	Study Title	Cell Target
NCT05215015	Early Phase 1	Recruiting	Study of Anti-CD33/CLL1 CAR-NK in Acute Myeloid Leukemia	CD33/CLL1
NCT05194709	Early Phase 1	Recruiting	Study of Anti-5T4 CAR-NK Cell Therapy in Advanced Solid Tumors	5T4
NCT05008536	Early Phase 1	Recruiting	Anti-BCMA CAR-NK Cell Therapy for the Relapsed or Refractory Multiple Myeloma	BCMA
NCT03692663	Early Phase 1	Recruiting	Study of Anti-PSMA CAR NK Cell (TABP EIC) in Metastatic Castration-Resistant Prostate Cancer	PSMA
NCT05248048	Early Phase 1	Recruiting	NKG2D CAR-T Cells to Treat Patients With Previously Treated Liver Metastatic Colorectal Cancer	NKG2D
NCT05247957	Phase 1	Recruiting	NKG2D CAR-NK Cell Therapy in Patients With Relapsed or Refractory Acute Myeloid Leukemia	NKG2DL
NCT05472558	Phase 1	Recruiting	Clinical Study of Cord Blood-derived CAR-NK Cells Targeting CD19 in the Treatment of Refractory/Relapsed B-cell NHL	CD19
NCT04887012	Phase 1	Recruiting	Clinical Study of HLA Haploidentical CAR-NK Cells Targeting CD19 in the Treatment of Refractory/Relapsed B-cell NHL	CD19
NCT05213195	Phase 1	Recruiting	NKG2D CAR-NK Cell Therapy in Patients With Refractory Metastatic Colorectal Cancer	NKG2D
NCT05008575	Phase 1	Recruiting	Anti-CD33 CAR NK Cells in the Treatment of Relapsed/Refractory Acute Myeloid Leukemia	CD33
NCT05507593	Phase 1	Recruiting	Study of DLL3-CAR-NK Cells in the Treatment of Extensive Stage Small Cell Lung Cancer	DLL3
NCT05410041	Phase 1	Recruiting	Anti-CD19 CAR-Engineered NK Cells in the Treatment of Relapsed/Refractory B-cell Malignancies	CD19
NCT04796675	Phase 1	Recruiting	Cord Blood Derived Anti-CD19 CAR-Engineered NK Cells for B Lymphoid Malignancies	CD19
NCT04623944	Phase 1	Recruiting	NKX101, Intravenous Allogeneic CAR NK Cells, in Adults With AML or MDS	NKG2D
NCT05020678	Phase 1	Recruiting	NKX019, Intravenous Allogeneic Chimeric Antigen Receptor Natural Killer Cells (CAR NK), in Adults With B-cell Cancers	CD19
NCT05563545	Phase 1	Recruiting	Anti-CD19 CAR-Engineered NK Cells in the Treatment of Relapsed/Refractory Acute Lymphoblastic Leukemia	CD19
NCT04796688	Phase 1	Recruiting	Universal Chimeric Antigen Receptor-modified AT19 Cells for CD19+ Relapsed/Refractory Hematological Malignancies	CD19
NCT05379647	Phase 1	Recruiting	Natural Killer (NK) Cell Therapy for B-Cell Malignancies	CD19
NCT05182073	Phase 1	Recruiting	FT576 in Subjects With Multiple Myeloma	-
NCT05410717	Phase 1/Phase 2	Recruiting	CLDN6-CAR-NK Cell Therapy for Advanced Solid Tumors	Claudin6
NCT05528341	Phase 1/Phase 2	Recruiting	NKG2D-CAR-NK92 Cells Immunotherapy for Solid Tumors	NKG2D
NCT03056339	Phase 1/Phase 2	Active, not recruiting	Umbilical & Cord Blood (CB) Derived CAR-Engineered NK Cells for B Lymphoid Malignancies	CD19
NCT04847466	Phase 2	Recruiting	Immunotherapy Combination: Irradiated PD-L1 CAR-NK Cells Plus Pembrolizumab Plus N-803 for Subjects With Recurrent/Metastatic Gastric or Head and Neck Cancer	-

Last updated on 29 September 2022 from clinicaltrials.gov.

**Table 4 cancers-15-00663-t004:** Study details of the CAR-M-based clinical study.

NCT Number	Stage	Status	Study Title	Cell Target
NCT04660929	Phase I	Recruiting	CAR-macrophages for the Treatment of HER2 Overexpressing Solid Tumors	HER2

Last updated on 29 September 2022 from clinicaltrials.gov.

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
