# Peer review of "CAR-T: What Is Next?"

_cancers, 2023, doi:10.3390/cancers15030663_

Round 1

Reviewer 1 Report

The authors have embarked on a herculean task of discussing the future of CAR-T cell therapy for heme malignancies and solid tumors as well as ways to overcome toxicity. The writing is generally clear and explanations are easy to understand. It also provides a good overview with some thoughtful comments. This is quite a comprehensive review and I wonder if focusing it a little more would make it easier to read.

1) The abstract does not quite describe all the review will cover (since a substantial portion of review describes future directions in solid tumors which are not mentioned in abstract). 

2) Table 1 is somewhat out of date as it doesn't include new second line indications of Yescarta and Breyanzi, the FL indication for Kymriah, or adult ALL indication for Tecartus.

3) I think in section 3.1, the antigen positive relapse is a little oversimplified. For example, in lymphoma patients can lose persistence but still have durable remissions. Also, what about completely refractory disease with no response?

4) In section 3.4.1, could you elaborate more on the BITE and CAR-T combinations (BITEs targeting alternative targets) and potentially give a more concrete example?

5) In section 3.4.2, perhaps describe more about bispecific CARs for example CD19/CD22 or CD19/CD20 and clinical trials or differences in bispecific CAR designs?

6) Section 4.1 is probably too long and don't need to go into detail about CRS/ICANS management as seems to be outside of scope of this review. Also, I think language about tocilizumab contraindicated in ICANS may be a little too strong - may soften to "generally not recommended" or something to that effect. If going to discuss this, would probably explain why tocilizumab not recommended in ICANS (IL6 receptor antibody, transient increased IL6 levels that can cross BBB while toci can't cross, etc.). However, this may all be outside of scope of this review so maybe just take all of the management out. 

7) Regarding strategies to mitigate CRS/ICANS in section 4.2, instead of spending time on standard of care management, could discuss briefly prophylaxis strategies (tocilizumab, steroids, anakinra, siltuximab, etc.) or earlier intervention. 

8)In section 4.2.2.1, if discussing Off/On switch, consider referencing PMID 33408186. 

9) In section 4.2.2.2, you discuss how safety switches should be able to target only cells that lead to toxicity - I think in general though, that has not been the case with safety switches and they unfortunately usually target all cells which is a drawback. Also consider citing PMID 33624095 which is clinical trial which activated safety switch. 

10) For section 4.3-4.4, probably would make it all one section and then have subsections discussing strategies of mitigating on target/off tumor toxicity. Would probably more explicitly state that bigger concern with on target/off tumor toxicity with novel CAR targets since generally can mitigate CD19 toxicities. 

11) In Section 5.1, I think you spend too much time talking about CD19, CD22, BCMA which is a little redundant. 

12) I wonder if detailing some solid tumor clinical trials in section 5.4 is outside the scope of this review. It just doesn't seem to be in place with the rest of the review which is focusing more on methods of improvement as opposed to describing clinical trial data.

13) In section 7.1, re: target antigen selection, probably wouldn't focus as much on CD19/BCMA and just discuss that need other targetes.

14) In section 7.3, I think the costs you list are outdated (2018 citation). 

Author Response

Thank you for your comments. The authors would like to highlight that the review article is intended to give an overall picture about the direction of CAR-T therapy research in hematology and solid tumors.

It is not intended to focus on any specific tumor type, area of research or the pricing structure of CAR-T. Rather it is to provide a reference for readers and researchers on ongoing and planned research in CAR-T.

Likewise, the authors would like to highlight that there have been 2 previous publications from one of the authors discussing different points for CAR-T therapy. The last was in 2021 in Cancers as well. The previous papers discussed different aspects in more detail e.g., mechanism of resistance and side effects. Hence the authors’ decision not to repeat the same information but rather to provide a top line summary of what has been published/discussed before.

With that overall plan of the article, the authors would like to address your comments.

  1. The abstract does not quite describe all the review will cover (since a substantial portion of review describes future directions in solid tumors which are not mentioned in abstract). 

Response: Thank you for your comment. The abstract was revisited to clarify the scope of the manuscript.

  1. Table 1 is somewhat out of date as it doesn't include new second line indications of Yescarta and Breyanzi, the FL indication for Kymriah, or adult ALL indication for Tecartus.

Thank you for your comment, this has been updated and the table is formatted to be easier for reading

  1. I think in section 3.1, the antigen positive relapse is a little oversimplified. For example, in lymphoma patients can lose persistence but still have durable remissions. Also, what about completely refractory disease with no response?

Thank you very much for your comment. This was covered in more detail in the previous publication Cancers, 2021, 13, 853. Accordingly, we decided not to repeat this topic again. It was mentioned here summarized just for completeness of the paper.

  1. In section 3.4.1, could you elaborate more on the BITE and CAR-T combinations (BITEs targeting alternative targets) and potentially give a more concrete example?

Thank you for your comment. The section was updated to make it clearer.

  1. In section 3.4.2, perhaps describe more about bispecific CARs for example CD19/CD22 or CD19/CD20 and clinical trials or differences in bispecific CAR designs?

Thank you very much for your comment, a few more sentences were added to this section including some details about the pre-clinical work.

  1. Section 4.1 is probably too long and don't need to go into detail about CRS/ICANS management as seems to be outside of scope of this review. Also, I think language about tocilizumab contraindicated in ICANS may be a little too strong - may soften to "generally not recommended" or something to that effect. If going to discuss this, would probably explain why tocilizumab not recommended in ICANS (IL6 receptor antibody, transient increased IL6 levels that can cross BBB while toci can't cross, etc.). However, this may all be outside of scope of this review so maybe just take all of the management out. 

Thank you for your comment. However, the authors disagree that this section is out of scope since we are covering different hurdles that face research and/or treatment with CAR-T. The management of CAR-T related adverse events still represents a major issue.

The section was written with some details as to cover the ASCO guidelines for the management of immune related AEs in patients treated with CAR-T therapy. This was not included in the previous publications as the guidelines were published after the previous reviews.

For the wording regarding tocizilumab being strong, this is what is understood from the ASCO guidelines:

The mainstay of treatment of ICANS is supportive care and corticosteroids. Tocilizumab does not resolve ICANS and may worsen it. Because of the possibility that tocilizumab may worsen neurotoxicity, the management of ICANS may take precedence over the management of low-grade CRS when the two occur simultaneously (JCO, 39:3978-3992, 2021)

Based on the wording from the ASCO guidelines it is understood that Tocilizumab is contraindicated for the treatment of neurotoxicity as it can worsen the symptoms which are written in the article.

  1. Regarding strategies to mitigate CRS/ICANS in section 4.2, instead of spending time on standard of care management, could discuss briefly prophylaxis strategies (tocilizumab, steroids, anakinra, siltuximab, etc.) or earlier intervention. 

Thank you for your feedback, however, the authors covered in the manuscript the side effects and management of it as they represent current challenges facing treatment with CAR-T. The release of the ASCO guidelines for the management of CAR-T AEs was an important topic to cover and use it as a source. It was published after the publication of the previous manuscript covering the management of AEs, hence the authors wanted to add to what was previously published while avoiding any repetition regarding this point. For covering the prophylaxis, as previously mentioned, the authors’ intent of the manuscript is to give an overview of different important points regarding CAR-T therapy, not to go into details of every point covered in the manuscript. The prophylaxis is considered out of scope of the manuscript.

  1. In section 4.2.2.1, if discussing Off/On switch, consider referencing PMID 33408186. 

Thank you very much for this. This was addressed and the reference provided was added.

  1. In section 4.2.2.2, you discuss how safety switches should be able to target only cells that lead to toxicity - I think in general though, that has not been the case with safety switches, and they unfortunately usually target all cells which is a drawback. Also consider citing PMID 33624095 which is clinical trial which activated safety switch. 

Thank you for referring us for the reporting of an ALL case in Phase 1/2 study and the management of ICANS. Reading through the published case and the management of ICANS is more descriptive of suicide genes due to the mention of inducible caspase-9 (iC9). It covers the ON-switch CARs and OFF-switch CARs, and therefore this part was changed as follows to make it clearer.

ON-switch CARs and OFF-switch CARs

Engineered CAR-T cells once infused into the patients, they start a chain of immunological reactions that kills the target cells. However, these reactions continue in an uncontrolled way leading to a set of immunological toxicities which leads to the development of different symptoms of CRS. The release of these inflammatory cytokines in an uncontrolled manner results in the affection of different organs with different severities. On-switch CAR is a design that separates the signal domain from the costimulatory domain. The T cell activation can be activated by adding heterodimerizing small molecules to promote the assembly of two fragmented CARs. Furthermore, the extent of cell activation can be modified by the dosage of the molecules [40]. Another design to reduce toxicity is the Off-switch CAR which is also named small molecule-assisted shutoff (SMASh)-CAR. SMASh-CAR includes a degron domain in the CAR structure, and it enables CAR degradation when protease inhibitors are given, thus downregulating T-cell activity. Oppositely, without the administration of a protease inhibitor, protease would cut off the target site to remove degron and leads to the expression of CAR [41].

  1. For section 4.3-4.4, probably would make it all one section and then have subsections discussing strategies of mitigating on target/off tumor toxicity. Would probably more explicitly state that bigger concern with on target/off tumor toxicity with novel CAR targets since generally can mitigate CD19 toxicities. 

Thank you very much for your comment about how to present these 2 sections, however the authors would like to keep it as is as it covers the points they would like to discuss in this review.

  1. In Section 5.1, I think you spend too much time talking about CD19, CD22, BCMA which is a little redundant. 

Thank you very much for your comment, however, we disagree that it is redundant and would like to keep it as it is written.

  1. I wonder if detailing some solid tumor clinical trials in section 5.4 is outside the scope of this review. It just doesn't seem to be in place with the rest of the review which is focusing more on methods of improvement as opposed to describing clinical trial data.

Thank you very much for your comment, however, based on what was mentioned at the beginning of this response the authors believe this is an important point to cover in the manuscript and would like to keep it as written.

  1. In section 7.1, re: target antigen selection, probably wouldn't focus as much on CD19/BCMA and just discuss that need other targets.

Thank you very much for your comment. However, the authors would like to keep this.

  1. In section 7.3, I think the costs you list are outdated (2018 citation). 

Thank you for your comment, however the manuscript is neither covering the changes of the cost of CAR-T therapy over time, nor any costing or pricing strategies. The intention was just to highlight the issue of CAR-T therapy being very expensive rendering it not available on a wide scale for all the patients who might benefit from it. Since the price is correctly reflective of what was published about the price in 2018, accordingly the authors would like to keep it.

Attached is the final manuscript after going through professional editing and language check.

Reviewer 2 Report

The authors Yi-Ju et al submitted a review article entitled “CAR-T, what is next?” for publication in MDPI-Cancers.

The article covers CAR-T cells and various modifications of the therapy, its uses, and disadvantages. The manuscript falls within the scope of MDPI-Cancers. However, I have given a few comments which should be addressed before publishing this manuscript.

The manuscript needs a thorough English editing service. Throughout the paper, the sentence formation is very odd and it seems that it was done deliberately to avoid plagiarism. But at the same time, the authors are requested to maintain the continuity/flow of the subject.

Line 21: “Solid” tumors…

Line 21: “therapy treatment”?

Line 58/59: Safety concerns? Please, write down what are the side effects?

Line 60: Now, the authors are mentioning CRS etc? This should be in continuation with the previous sentence. The authors are requested to maintain continuity so that the readers can understand easily.

Line 69: Pathophysiologic.

Line 84: Dependent.

Line 91: Hematological malignancies or cancer but “Hematological tumors” is not used generally.

Line103: of

Line110: resulting in a.

Line114: were.

Line 117: that resulted.

Line 121: In Figure 2: the left image is not named?  What is the image of? Is the general CAR structure…if yes mention it in the figure also.

Line 145: “membrane of CD19”?

Line  149: sentence to be rephrased. Difficult to understand and grammatical error.

Line 185: different hematological malignancies and then mentioning only ALL?

Line 186: not sure what the authors imply by this sentence? Very vague?

Line 202: are not is.

Line 255: thus compromising efficacy? Not clear what the authors imply here?

Line 262: affection? Not generally used in this sense so please change the word or rephrase the sentence.

Line 277/278: However, cellular therapies can 277 pose significant? Incomplete sentence?

Line 283: 2? Two.

Line 293: 4.3. On-target, Off-tumor activity: First few lines are okay but then the flow is missing. The authors jump from one sentence to another sentence without a proper conclusion or with a finishing sentence. Please, rephrase this.

Line 305:4.4. Novel CAR designs to avoid on-target, off-tumor activity: Not sure what the authors are implying here or what the concept is. It seems that the authors are not clear on the concept or at least not able to put it properly. Should rewrite this. What does it mean by logic gates? First, this has to be explained? Not all readers would understand. As the subsequent paragraphs are based on similar concept, please explain.

General: The authors write “tumor” sometimes and “tumour” sometimes. The authors should stick to one of them.

Line 345: hematological tumor? Malignancy/cancer.

Line 371: sending…injecting?

Line 398: Overly?  “Overlay”.

Line 430: PR and PSA. Please, expand when you introduce the first time.

Line 441: Here you can write PR once you introduce this in Line 430.

Line 442: PFS introduced for the first time has to be expanded.

Line 458: OS introduced for the first time has to be expanded.

Line 539: which has not who has

Line 559: have an alike expression?

Line 601:  have made it achieved?

Author Response

Thank you for your comments. All were addressed. The manuscript was submitted to the editing department for review.

Line 21: “Solid” tumors…

Corrected.

Line 21: “therapy treatment”?

Corrected and clarified.

Line 58/59: Safety concerns? Please, write down what are the side effects?

This was addressed, currently it reads: the safety concern was another problem for the patients as well as treating physicians e.g., CRS and neurologic toxicities.

Line 60: Now, the authors are mentioning CRS etc? This should be in continuation with the previous sentence. The authors are requested to maintain continuity so that the readers can understand easily.

With the addition of some of the examples of toxicity now the sentence is clear.

Line 69: Pathophysiologic.

This was corrected.

Line 84: Dependent.

Thank you this was corrected.

Line 91: Hematological malignancies or cancer but “Hematological tumors” is not used generally.

Thank you. This was corrected to malignancies.

Line103: of

This was corrected.

Line110: resulting in a.

This was corrected.

Line114: were.

This was corrected.

Line 117: that resulted.

Corrected.

Line 121: In Figure 2: the left image is not named?  What is the image of? Is the general CAR structure…if yes mention it in the figure also.

It is mentioned in the ligand below the graph. The authors will not be able to make changes to the PDF

Line 145: “membrane of CD19”?

Thank you, corrected now it reads epitope of CD19.

Line 149: sentence to be rephrased. Difficult to understand and grammatical error.

Thank you for your feedback, the whole sentences were revised. Now it reads:

Relapse after initial response to CAR-T cell is still a major problem.  Different strategies were developed to overcome this issue e.g., the development of novel CAR-T designs and combinations with other therapies.

Line 185: different hematological malignancies and then mentioning only ALL?

Thank you for your comment. Based on the review from the other reviewer, the whole paragraph was revised to include information about CAR-T.BiTE .

Line 186: not sure what the authors imply by this sentence? Very vague?

As mentioned above the whole paragraph was rephrased.

Line 202: are not is.

Thank you, corrected.

Line 255: thus compromising efficacy? Not clear what the authors imply here?

Thank you. Now this reads:

that could lead to decrease efficacy of the CAR-T therapy

Line 262: affection? Not generally used in this sense so please change the word or rephrase the sentence.

Thank you for your comment. This is now corrected so it reads:

The release of these inflammatory cytokines in an uncontrolled manner impacts different organs with different severities.

Line 277/278: However, cellular therapies can 277 pose significant? Incomplete sentence?

Thank you very much. This was removed.

Line 283: 2? Two.

Thank you very much. This was changed.

Line 293: 4.3. On-target, Off-tumor activity: First few lines are okay but then the flow is missing. The authors jump from one sentence to another sentence without a proper conclusion or with a finishing sentence. Please, rephrase this.

Thank you very much, this is now changed to: An ideal target for CAR-T should only exist on the cancer cells and be expressed at a high level. Unfortunately, most antigens are present on both tumour cells and normal cells resulting in CAR-T cells therapy targeting antigens on both normal and cancer cells. When CAR-T cells attack antigens that coexist in some normal tissues, this is what is called ‘on-target, off-tumor’ toxicity.

Line 305:4.4. Novel CAR designs to avoid on-target, off-tumor activity: Not sure what the authors are implying here or what the concept is. It seems that the authors are not clear on the concept or at least not able to put it properly. Should rewrite this. What does it mean by logic gates? First, this has to be explained? Not all readers would understand. As the subsequent paragraphs are based on similar concept, please explain.

The authors would like to express their reservation on the comment that the authors are not clear on the concept or not able to put it properly. There has been some text that was inadvertently deleted.

The whole paragraph was changed to clarify the meaning of gated or logic-gated approach.

In Mice studies, CAR-T cells targeting tumors with ROR1 antigen following lymphodepletion resulted in lethal bone marrow failure. This is due to the targeting of ROR1 expressed on normal stromal cells. The ability to develop a CAR-T that selectively targets antigens present on tumor cells and not normal cells will improve the efficacy results but avoid severe toxicities-the so called gated or logic-gated approach. From pre-clinical studies synNotch receptors induced transcriptional activation in response to combinatorial target-antigen recognition. However, data from clinical studies are needed to define the applicability of different strategies in clinical practice [31].

General: The authors write “tumor” sometimes and “tumour” sometimes. The authors should stick to one of them.

This occurred due to the presence of the authors in the UK and North America. The difference is due to the difference in writing the word in British vs American English. All “tumour” words were changed to “tumor:”

Line 345: hematological tumor? Malignancy/cancer.

Changed to malignancy.

Line 371: sending…injecting?

The sentence was changed and currently it reads: CAR-T cells are directly injected into the tumor site.

Line 398: Overly?  “Overlay”.

Thank you for your query. Overly is correct. It means according to Oxford and Webster dictionaries: excessive/ to an excessive degree.

Line 430: PR and PSA. Please, expand when you introduce the first time.

Thank you very much this was addressed.

Line 441: Here you can write PR once you introduce this in Line 430.

Thank you very much. Change implemented.

Line 442: PFS introduced for the first time has to be expanded.

Done.

Line 458: OS introduced for the first time has to be expanded.

Done

Line 539: which has not who has

Done.

Line 559: have an alike expression?

Thank you very much, this has changed to “similar expression”

Line 601:  have made it achieved?

Thank you very much. The whole sentence was changed to:

The notable outcomes of CAR-T cell therapy were considered a clinical success, however on the commercial side, CAR-T therapy has achieved minimal success.

Attached below is the final manuscript after review by the editing department.

Round 2

Reviewer 1 Report

I just have a few remaining comments. 

1) I couldn't find the updated Table 1 with new indications for products. 

2) Regarding section 4.2.2.2 and discussion of suicide genes, regarding lines 303-308, usually when suicide genes are activated (as in the case of AP1903 and iCasp9), this involves the non-selective apoptosis of CAR-T cells. Just reading through the paragraph, I interpreted it as saying that only cells leading to toxicity are targeted. 

Author Response

Reply to the reviewer: 

  • Thank you very much for your comments. We apologize for the mistake in the table. We had several drafts going on so that led to this oversight. The corrected one is in now in the updated draft of the manuscript.
  •  For the comment regarding the suicide genes. You have kindly mentioned they are in lines 303-308. However, we believe the lines that discuss suicide genes are 267-283.

The authors agree with your comment, we have redacted a couple of sentences in the paragraph as shown below:

CAR-T therapy is currently the most effective therapy for its approved indications, but the cellular killing ability of the cells is not restricted to cancer cells only. It can affect normal cells leading to off-target effects and different toxicities. To minimize these toxicities, a genetically encoded molecule aiming at the selective destruction of the transferred cells was developed. This is called a suicide gene. The addition of a suicide gene with CAR-T cells leads to the selective destruction of the adoptively transferred cells as well as a selective ablation of the genetically modified cells, preventing the off-target effect of CAR-T therapy and the toxicity of the therapy. A suicide gene needs only to be able to target cells that lead to toxicity, irreversibly eliminating them without impacting the cells that are killing the cancer cells, hence leading to CAR-T efficacy. There are several validated suicide genes, two of which are among the most studied in the clinical setting: herpes-simplex-thymidine-kinase (HSV-TK) and inducible-caspase-9 (iCasp9). The iCasp9 suicide gene can trigger apoptosis rapidly when exposed to dimerizing agents (AP1903), thereby leading to an irreversible termination of T cells, and the inducible HSV-TK suicide gene is activated by the administration of ganciclovir (GCV) [41]. Similarly, the strategy of suicide receptors was pursued; it depends on inducing another antigen (e.g., CD20) expression on the surface of CAR-T cells, thereby serving as a "suicide receptor" that, when anti-CD20/rituximab is infused, will bind to the receptors and eliminate CAR-T cells [39, 40, 42].

Currently the paragraph reads:

CAR-T therapy is currently the most effective therapy for its approved indications, but the cellular killing ability of the cells is not restricted to cancer cells only. It can affect normal cells leading to off-target effects and different toxicities. To minimize these toxicities, a genetically encoded molecule aiming at the selective destruction of the transferred cells was developed. This is called a suicide gene. The addition of a suicide gene with CAR-T cells leads to the selective destruction of the adoptively transferred cells as well as a selective ablation of the genetically modified cells, preventing the off-target effect of CAR-T therapy and the toxicity of the therapy. There are several validated suicide genes, two of which are among the most studied in the clinical setting: herpes-simplex-thymidine-kinase (HSV-TK) and inducible-caspase-9 (iCasp9). The iCasp9 suicide gene can trigger apoptosis rapidly when exposed to dimerizing agents (AP1903), thereby leading to an irreversible termination of T cells, and the inducible HSV-TK suicide gene is activated by the administration of ganciclovir (GCV) [41]. Similarly, the strategy of suicide receptors was pursued; it depends on inducing another antigen (e.g., CD20) expression on the surface of CAR-T cells, thereby serving as a "suicide receptor" that, when anti-CD20/rituximab is infused, will bind to the receptors and eliminate CAR-T cells [39, 40, 42].

We hope this addresses all your comments on the manuscript. We sincerely thank you for your thorough review and important feedback.

Attached below the updated draft of the manuscript.
